# Simulation of miniscrew-root distance available for molar distalization depending on the miniscrew insertion angle and vertical facial type

Ju-Hyun Yoon[1], Jung-Yul Cha[1], Yoon Jeong Choi●[1], Won-Se Park[2], Sang-Sun Han[3], Kee-Joon Lee●[1]*

1 Department of Orthodontics, Institute of Cranio-Facial Deformity, College of Dentistry, Yonsei University, Seoul, Korea, 2 Department of Advanced General Dentistry, College of Dentistry, Yonsei University, Seoul, Korea, 3 Department of Oral and Maxillofacial Radiology, College of Dentistry, Yonsei University, Seoul, Korea

* orthojn@yuhs.ac

**Data Availability Statement:** Data cannot be made publicly available as the data contain sensitive and identifying information. The authors confirm that the data will be made available upon request.

## Abstract

### Objective

The purpose of this study was to evaluate the effects of miniscrew insertion angle and vertical facial type on the interradicular miniscrew–root distance available for molar distalization.

### Materials and methods

Cone-beam computed tomography images of 60 adults with skeletal Class I occlusion exhibiting hyperdivergent (n = 20), normodivergent (n = 20), and hypodivergent (n = 20) facial types were used. Placement of a 6-mm long, 1.5-mm diameter, tapered miniscrew was simulated at a site 4 mm apical to the cementoenamel junction, with insertion angles of 0˚, 30˚, 45˚, and 60˚ relative to the transverse occlusal plane. The shortest linear distance between the miniscrew and anterior root at four interradicular sites was measured: maxillary second premolar and first molar (Mx 5–6), maxillary first and second molars (Mx 6–7), mandibular second premolar and first molar (Mn 5–6), and mandibular first and second molars (Mn 6–7).

### Results

Miniscrew–root distance significantly increased as the insertion angle increased from 0˚ to 60˚. In the mandible, the distances significantly differed among vertical facial types, increasing in the following order: hyperdivergent, normodivergent, and hypodivergent. The minimum mean distance was found in the Mx 6–7 (30˚; 0.86±0.35 mm), and the maximum mean distance was found in the Mn 5–6 (60˚; 2.64±0.56 mm). The rates of miniscrews located buccally outside the root distalization path were up to 70% and 55% when the miniscrews were placed at 60˚ insertion angles in the Mx 5–6 and Mn 5–6 regions, respectively.

Requests may be sent to the corresponding author (orthojn@yuhs.ac). For non-author institutional contact for data access, data are available from the institutional review board of Yonsei University Dental Hospital (contact via dentalirb@yuhs.ac) for researchers who meet the criteria for access to confidential data.

**Funding:** This study was supported by faculty research grant of Yonsei University College of Dentistry, Grant/Award Number:2017-0012 The funders had no role in study design, data collection and analysis, decision to publish, or preparation of the manuscript.

**Competing interests:** The authors have declared that no competing interests exist.

## Conclusions

Miniscrew–root distance increased significantly with the increased insertion angle, and the amount of increase was affected by the miniscrew placement site and vertical facial type. To ensure adequate distalization of the posterior segment, the miniscrew should be inserted at an angle in the interradicular area between the second premolar and first molar.

## Introduction

Miniscrews provide effective anchorage for molar distalization and are indicated for non-extraction treatments. It has been shown that maxillary incisors and molars can be simultaneously moved distally without loss of incisor anchorage, unlike with the conventional pendulum and distal jet [1, 2]. With respect to insertion sites, either the buccal interradicular area or palatal side has been proposed. In contrast to bone-borne palatal appliances, which often cause patient discomfort because of their complex structures, the buccal interradicular miniscrew is simple and reduces patient discomfort because the elastic chains are engaged directly on the archwire. However, extensive molar distalizations are difficult with buccal interradicular miniscrews because they can come in contact with the anterior roots with the distal movement of the teeth [2]. It has been reported that contact between the miniscrew and the root can cause miniscrew failure [3, 4].

Several studies have recommended oblique, rather than perpendicular, insertions of the buccal interradicular miniscrews [5–7]. Park [8] and Park et al. [9] recommended oblique insertions of miniscrews at angles of 30–40˚ in the maxilla and 10–20˚ in the mandible to prevent root damage. When the miniscrew is placed obliquely, the tip of the miniscrew is directed toward the apical portion of the roots, where the interdental space is wider [10], reducing the depth of penetration of the miniscrew into the buccal bone. As a result, the distance between the miniscrew and the root is increased, and molar distalization is rendered more favorable by the angulation of the miniscrew. However, these speculations have not been verified by research.

When estimating the miniscrew–root distance, buccal bone thickness with interradicular distance should be considered. Alveolar ridge thickness, which is known to be related to facial divergence, affects the distance between the miniscrew and the root in the buccolingual axis [11]. Previous studies on facial divergence associated with miniscrew use have compared the success rates of miniscrews among vertical facial types [12]; however, no study has compared the miniscrew–root distance among vertical facial types.

Various in-vitro studies have investigated the effect of miniscrew angulation on primary stability [13–16]. However, the effect of the miniscrew placement angle on the miniscrew–root distance has not yet been studied. In-vivo measurements of the miniscrew–root distances after the insertion of miniscrews at various angles at the same insertion point are difficult. In such cases, three-dimensional imaging simulation programs can analyze the effect of the miniscrew placement angles on root proximity, under control of other factors.

The purpose of this study was to investigate the effects of the miniscrew insertion angle and vertical facial type on miniscrew–root distances available for molar distalization in the maxilla and mandible via simulated placement of interradicular miniscrews using patients' cone-beam computed tomography (CBCT) images.

## Materials and methods

### Study group

The sample used in this simulation study was selected by screening the CBCT images of 60 patients who visited Yonsei University Dental Hospital from January 2016 to February 2017. The inclusion criteria were a skeletal and dental Class I (premolar and molar) relationship, with minimal (< 2 mm) posterior crowding in both arches. The exclusion criteria were as follows: 1) severe skeletal or dental asymmetry, 2) posterior arch discrepancy, 3) severe root dilacerations or excessively short roots, 4) periodontal disease with vertical alveolar bone loss, 5) missing teeth (except for the third molars), 6) presence of prostheses, 7) severe sinus pneumatization, and 8) history of orthodontic treatment. Ethical approval was obtained from the institutional review board of Yonsei University Dental Hospital (approval number: 2-2018-0014). CBCT scans (Alphard 3030; Asahi Roentgen Inc., Kyoto, Japan) were performed to assess the presence and conditions of impacted third molars, supernumerary teeth, and/or other pathologies (e.g., cysts). The images were acquired in a single 360˚ rotation with a scan time of 17 seconds, 80 kVp, 10 mA, 0.39 mm voxel size, and a field of view of $20.0 \times 20.0$ cm.

Two-dimensional cephalometric images derived from the CBCT scans were used to classify CBCT images into one of three vertical facial groups. Patients were classified into hyperdivergent, normodivergent, or hypodivergent groups using one angular (S-N/Go-Me) and one linear (S-Go/N-Me) measurement. An S-N/Go-Me angle < 27˚ indicated hypodivergence, between 27˚ and 37˚ indicated normodivergence, and > 37˚ indicated hyperdivergence [17]. An S-Go/N-Me ratio < 61% indicated hyperdivergence, between 61% and 69% indicated normodivergence, and > 69% indicated hypodivergence [18]. If angular and linear measurements indicated different group assignments for a particular patient, those images were excluded from subsequent analyses. A total of 60 CBCT images were obtained by applying all inclusion and exclusion criteria such that each facial group consisted of 20 patients. Since our study is an explorative pilot study, 20 subjects for each group were determined in consideration of sample numbers suggested as suitable for the pilot study [19–21]. The total sample included 23 men and 37 women, and the average age was $26.2 \pm 7.7$ years (age range, 20–47 years). Patient characteristics are listed in Table 1.

### Interradicular miniscrew insertions and miniscrew–root distance measurements

All CBCT images were first saved as Digital Imaging and Communication in Medicine files (slice thickness: 1.0 mm) and then reconstructed into three-dimensional images using the InVivo Dental software (version 5.4; Anatomage, San Jose, CA, USA).

**Table 1. Patient characteristics.**

| Variable | Hyperdivergent[a] | Normodivergent[b] | Hypodivergent[c] | p value |
|---|---|---|---|---|
| Age (y) | 25.3±6.1 | 25.2±7.2 | 28.2±8.9 | .213† |
| Sex(M/F) | 4/16 | 7/13 | 12/8 | .032‡ |
| Mandibular plane angle (S-N/Go-Me)(˚) | 42.5±3.7 | 33.6±2.3 | 23.9±3.7 | <.001† |
| | | | | a>b>c |
| Facial height index (S-Go/N-Me)(%) | 58.7±1.9 | 66.8±1.8 | 75.3±3.5 | <.001† |
| | | | | c>b>a |

S, sella; N, nasion; Go, gonion; Me, menton.

†one-way ANOVA.

‡Chi-square test.

The reconstructed three-dimensional images were reoriented for performing measurements across three planes without inducing any measurement errors caused by non-standardized head postures. The anatomic occlusal plane was aligned parallel to the horizontal axis of the software in the sagittal view. The transverse occlusal line connecting the mesiobuccal cusps of the maxillary first molars were aligned parallel to the horizontal axis of the software in the coronal view, and the line connecting the mesiobuccal cusps of the maxillary first molars were aligned parallel to the horizontal axis of the software in the axial view (Fig 1).

A simulated insertion of a miniscrew of a desired type, diameter, and length can be performed on reconstructed CBCT images using the InVivo Dental software. In this study, a tapered miniscrew of 1.5 mm diameter and 6 mm length was selected, as recommended in a previous study (Fig 2) [22]. Miniscrews were inserted in the maxillary and mandibular buccal alveolar bone at four interradicular sites: between the maxillary second premolar and first molar (Mx 5–6), between the maxillary first and second molars (Mx 6–7), between the mandibular second premolar and first molar (Mn 5–6), and between the mandibular first and second molars (Mn 6–7). The insertion point was located 4 mm apical to the cementoenamel junction (CEJ) of the adjacent teeth.

For the placement of the miniscrew, the long axis of the miniscrew was positioned parallel to the horizontal axis of the software in the coronal view. The head of the miniscrew was in contact with the cortical bone, and all threaded portions were placed in bone. The miniscrew was verified in the axial and sagittal views to be placed at the mesiodistal midpoint between the roots (Fig 3). The miniscrew was then vertically angulated at 0˚, 30˚, 45˚, and 60˚ relative to the horizontal axis of the software, which is parallel to the transverse occlusal plane, while maintaining the position of the midpoint between the roots (Fig 4).

Two points (apex and neck) on the long axis of the miniscrew were determined on the coronal view for measuring the miniscrew–root distance. The distance was measured on serial CBCT axial images obtained by slicing the CBCT images in the axial plane at 0.1 mm slice thicknesses between the apex and the neck. The line connecting the buccal cusps of the first and second molars was used as a reference. In each axial plane, the shortest linear distance from the mesial surface of the miniscrew to the distal surface of the root of the anterior tooth

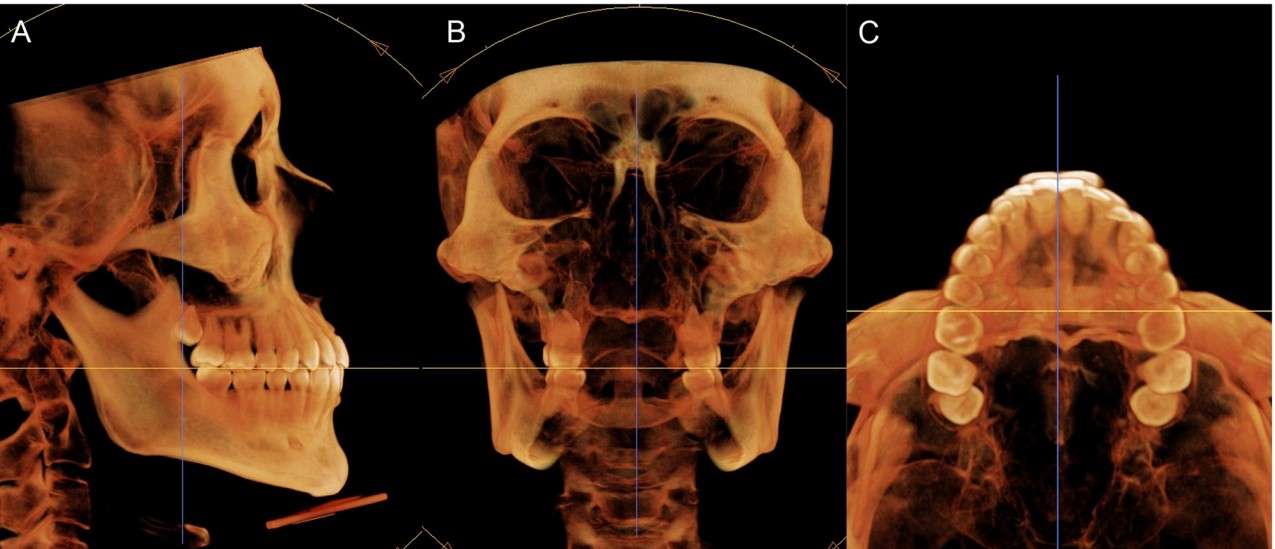

**Fig 1. Image re-orientation using the InVivo Dental software.** (A) In the sagittal view, the images are oriented using the occlusal plane as a reference. (B, C) In the frontal and axial views, the images are oriented referring to a line passing through the mesiobuccal cusps of maxillary first molars.

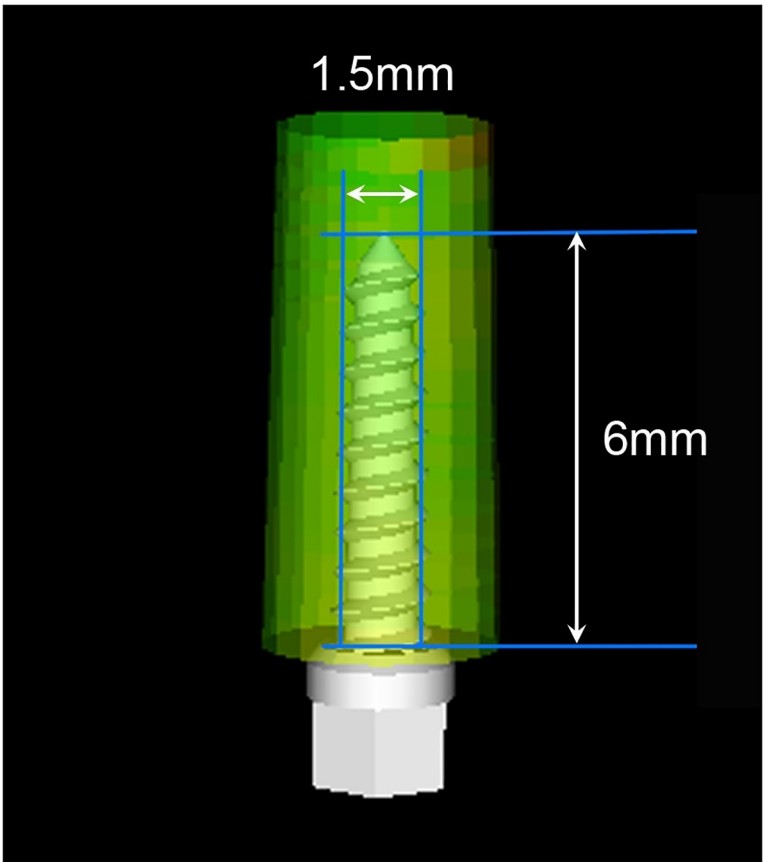

**Fig 2. Schematic diagram of a miniscrew.** Tapered miniscrew with 1.5 mm diameter and 6 mm length is reconstructed by the InVivo dental software.

was measured on a line parallel to the reference line, and the smallest measured value between the apex and the neck was recorded as the miniscrew–root distance (Fig 5).

## Statistical analyses

All statistical analyses were performed using SPSS for Windows (version 20.0; SPSS Inc, Chicago, IL, USA). As paired t-tests revealed no statistically significant differences between measurements acquired from the left and right sides, the average of the bilateral measurements was used. All measurements were performed by the same examiner. Measurements for five samples of each facial type were repeated after 2 weeks to test the intra-examiner reliability. Intraclass correlations revealed statistically significant reliability (ICC [Intraclass Correlation Coefficient] = 0.99).

In two cases, the miniscrew–root distances were categorized as non-measurable: a contact group, in which the miniscrews were directly in contact with the roots on miniscrew insertion, and a noncontact group, in which the miniscrews did not interfere with root movements because they were placed buccally outside the path of root distalization. Therefore, in descriptive statistics, means and standard deviations were calculated using only measurable values of miniscrew–root distances, and rates of non-measurable cases (contact group, noncontact group) were determined for each insertion site.

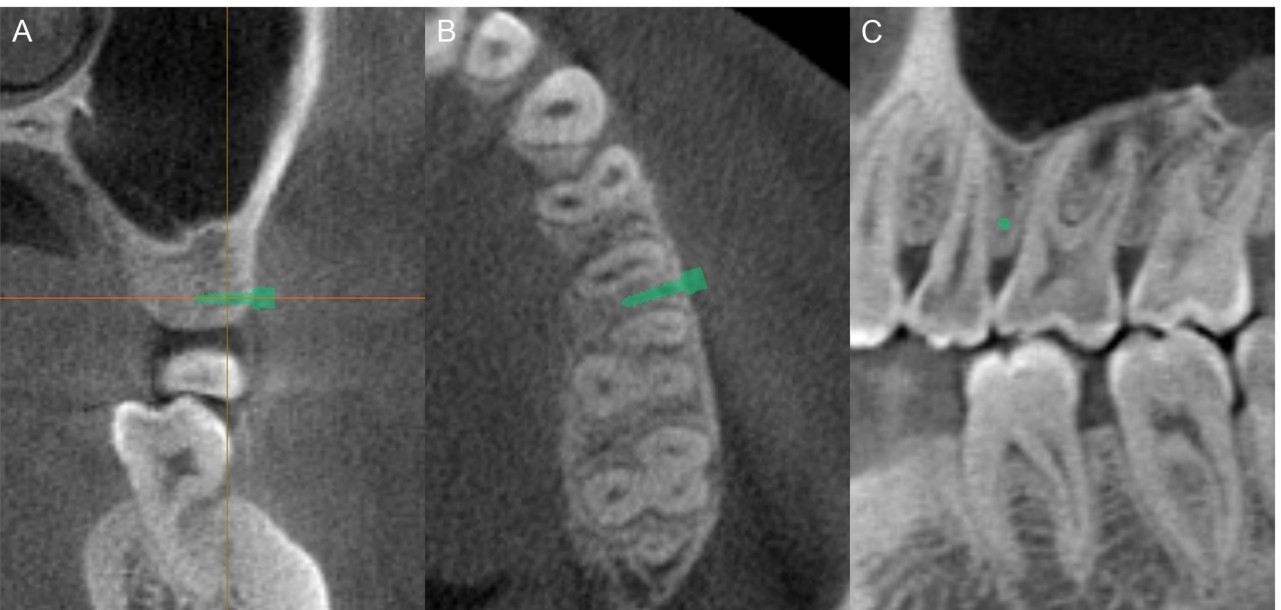

**Fig 3. Miniscrew placement.** (A) The long axis of the miniscrew should be parallel to the horizontal axis of the software. (B) The miniscrew is then placed at the mesiodistal midpoint between the roots in the axial view. (C) The miniscrew is seen as a circular dot in the sagittal view.

For statistical analyses, all values of the miniscrew–root distance, including those of the non-measurable cases, were converted to categorical variables and were assigned values between 0 and 6 (Table 2). Generalized estimating equations (GEE) were formulated using ordinal logistic modeling and were used to determine differences among categorical values. Sex, mandibular plane angle, and facial height index, which differed significantly among groups (Table 1), were used as covariates.

## Results

Means and standard deviations for measurable miniscrew–root distances at each insertion site are shown in Table 3. The average miniscrew–root distance at Mx 5–6, Mx 6–7, Mn 5–6, and

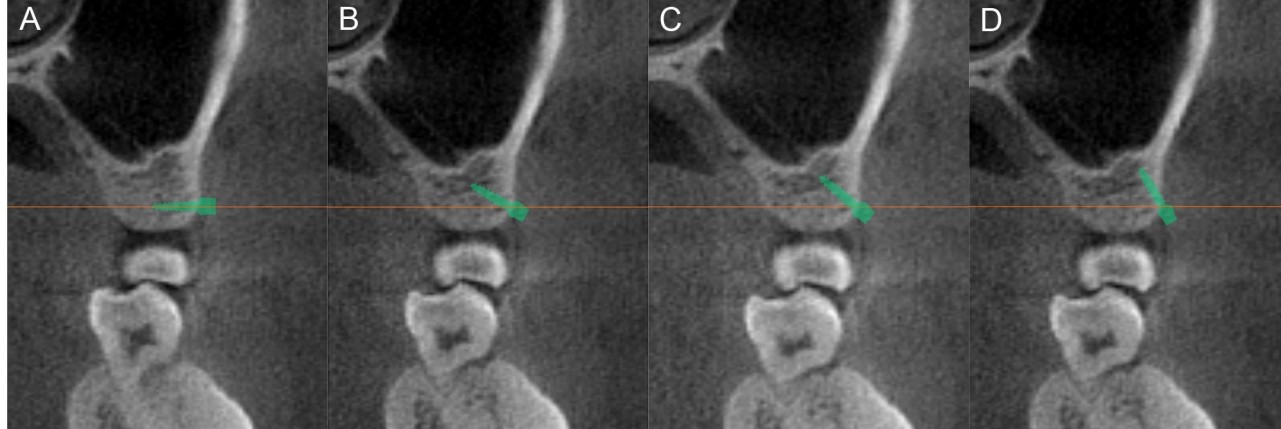

**Fig 4. Miniscrew insertion angles.** The miniscrews are placed at four different vertical angles relative to the horizontal axis of the software, which is parallel to the transverse occlusal plane. (A) 0˚; (B) 30˚; (C) 45˚; (D) 90˚.

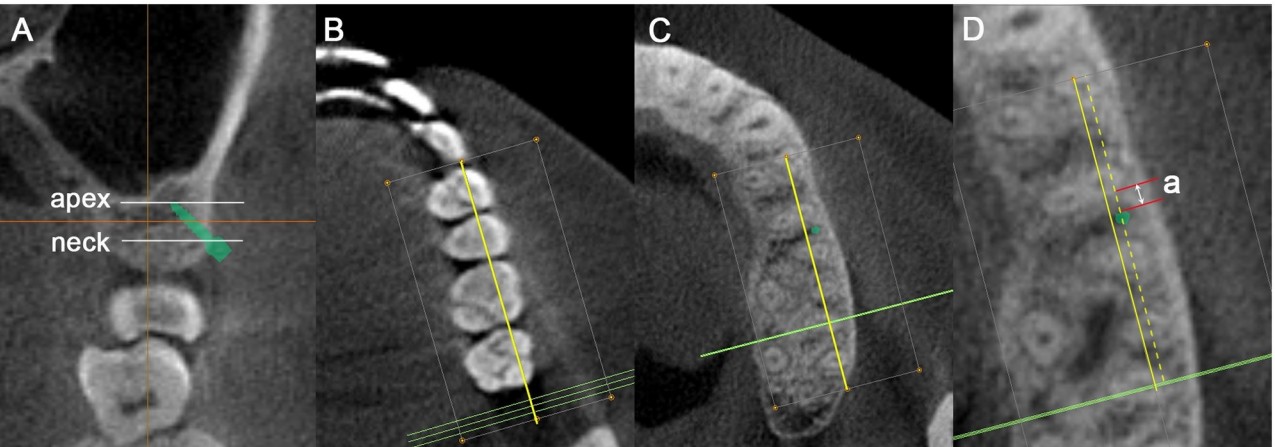

**Fig 5. Measurement of miniscrew–root distance.** (A) Apex and neck of the miniscrew in the coronal view. (B) Axial slice at occlusal level: yellow line, reference line connecting the buccal cusps of first and second molars. (C) Axial slice at root level: green circular dot, axial cross section of the angulated miniscrew. (D) Magnified view of (C): dotted yellow line, line parallel to the reference line; a, shortest miniscrew–root distance.

Mn 6–7 ranged from 1.08 to 2.16 mm, 0.86 to 1.17 mm, 1.04 to 2.64 mm, and 1.03 to 2.05 mm, respectively. At Mx 5–6, Mn 5–6, and Mn 6–7 miniscrew placement sites, the minimum mean of the measurable miniscrew-root distance (Mx 5–6: 1.08±0.25 mm; Mn 5–6: 1.04±0.14 mm; Mn 6–7: 1.03±0.25 mm) was measured at a 0˚ insertion angle in hyperdivergent facial types, and the maximum mean of the measurable miniscrew-root distance (Mx 5–6: 2.16±0.33 mm; Mn 5–6: 2.64±0.56 mm; Mn 6–7: 2.05±0.61 mm) was measured at a 60˚ insertion angle in hypodivergent facial types. In exception, in the Mx 6–7 region, the minimum mean of the measurable miniscrew-root distance (Mx 6–7: 0.86±0.35 mm) was measured at a 30˚ insertion angle in hyperdivergent facial types, and the maximum mean of the measurable miniscrew-root distance (Mx 6–7: 1.17±0.48 mm) was measured at a 60˚ insertion angle in hypodivergent facial types (Table 3).

The rate of miniscrew–root distances > 2 mm were calculated in case of miniscrews placed at angles of 60˚. The rates of Mx 5–6 and Mn 5–6 were 71.7% and 78.3%, respectively; the corresponding values of Mx 6–7 and Mn 6–7 were 15% and 25%, respectively (Table 4).

**Table 2. Categorization of miniscrew–root distance: Measurable values and non-measurable values (contact group, noncontact group).**

| Miniscrew–root distance (mm) | categorization | count |
|---|---|---|
| Contact group | 0 | 26 |
| 0 < distance ≤0.5 | 1 | 9 |
| 0.5 < distance ≤1 | 2 | 163 |
| 1 < distance ≤1.5 | 3 | 395 |
| 1.5 < distance ≤2 | 4 | 195 |
| 2 < distance | 5 | 104 |
| Noncontact group | 6 | 68 |
| P | | <.001‡ |

Contact group: The miniscrew directly contacts the root upon miniscrew insertion.

Noncontact group: The miniscrew does not interfere with root movement because it is located buccally outside the root distalization path.

‡Chi-square test.

**Table 3. The means and standard deviations of measurable miniscrew–root distance values (unit: mm).**

| Jaw | Site | Angle | Total | Hyper | Normo | Hypo |
|---|---|---|---|---|---|---|
| Mx | 5–6 | 0˚ | 1.14±0.23 | 1.08±0.25 | 1.15±0.20 | 1.19±0.22 |
| | | 30˚ | 1.50±0.40 | 1.37±0.34 | 1.46±0.44 | 1.67±0.38 |
| | | 45˚ | 1.73±0.37 | 1.62±0.34 | 1.69±0.36 | 1.90±0.37 |
| | | 60˚ | 1.91±0.43 | 1.84±0.57 | 1.84±0.28 | 2.16±0.33 |
| | 6–7 | 0˚ | 0.98±0.23 | 0.96±0.26 | 0.97±0.20 | 1.01±0.25 |
| | | 30˚ | 0.91±0.29 | 0.86±0.35 | 0.93±0.23 | 0.94±0.29 |
| | | 45˚ | 0.99±0.33 | 0.95±0.32 | 0.91±0.23 | 1.09±0.40 |
| | | 60˚ | 1.12±0.39 | 1.17±0.41 | 1.03±0.27 | 1.17±0.48 |
| Mn | 5–6 | 0˚ | 1.18±0.19 | 1.04±0.14 | 1.20±0.15 | 1.31±0.17 |
| | | 30˚ | 1.46±0.25 | 1.24±0.17 | 1.49±0.18 | 1.65±0.21 |
| | | 45˚ | 1.76±0.40 | 1.41±0.24 | 1.80±0.22 | 2.09±0.38 |
| | | 60˚ | 2.18±0.50 | 1.91±0.38 | 2.25±0.41 | 2.64±0.56 |
| | 6–7 | 0˚ | 1.18±0.26 | 1.03±0.25 | 1.21±0.20 | 1.32±0.27 |
| | | 30˚ | 1.35±0.38 | 1.11±0.33 | 1.37±0.27 | 1.57±0.4 |
| | | 45˚ | 1.50±0.49 | 1.21±0.39 | 1.47±0.31 | 1.84±0.54 |
| | | 60˚ | 1.66±0.55 | 1.27±0.35 | 1.69±0.41 | 2.05±0.61 |

Mx, maxilla; Mn, mandible; 5–6, interradicular site between second premolar and first molar; 6–7, interradicular site between first and second molars; Hyper, hyperdivergent; Normo, normodivergent; Hypo, hypodivergent.

Root contact occurred only in the Mx 6–7 region. The rate of root contact varied according to the miniscrew insertion angle and ranged from 0 to 25% (Table 5).

Cases that could be assigned to the noncontact group were found at all miniscrew placement sites, and the rates ranged from 0 to 70%. The highest rate was observed between the second premolar and first molar, 60˚ insertion angle, and in the hypodivergent facial type, with 70% in the maxilla and 55% in the mandible (Table 5).

GEE analysis showed that the miniscrew–root distance significantly differed with facial type (p = 0.021), jaw (p<0.001), interradicular site (p<0.001), and insertion angle (p<0.001). Among vertical facial types, the miniscrew–root distance increased from the hyperdivergent to the normodivergent, and finally to the hypodivergent facial type. As for placement site, the miniscrew–root distance was greater in the mandible than that in the maxilla, and the distance was greater in the interradicular site between the second premolar and first molar compared with the site between the first and second molars. The miniscrew–root distance increased as the miniscrew placement angle increased (Table 6). The interaction between vertical facial type and jaw was also significant (p = 0.016). In the mandible, there were significant differences among vertical facial types, with the miniscrew–root distance increasing from

**Table 4. Rates of miniscrew–root distance > 2 mm (unit: %).**

| | 0˚ | 30˚ | 45˚ | 60˚ |
|---|---|---|---|---|
| Mx 5–6 | 0 | 10 | 30 | 71.7 |
| Mx 6–7 | 0 | 0 | 1.7 | 15 |
| Mn 5–6 | 0 | 0 | 31.7 | 78.3 |
| Mn 6–7 | 1.7 | 8.3 | 13.3 | 25 |

Mx, maxilla; Mn, mandible; 5–6, interradicular site between second premolar and first molar; 6–7, interradicular site between first and second molars.

**Table 5. The rates of non-measurable cases (contact group, noncontact group) at each miniscrew placement site and angle (unit: %).**

| jaw | Site | angle | Contact group (%) | | | Noncontact group (%) | | |
|---|---|---|---|---|---|---|---|---|
| | | | hyper | normo | hypo | hyper | normo | hypo |
| Mx | 5–6 | 0˚ | 0 | 0 | 0 | 0 | 0 | 0 |
| | | 30˚ | 0 | 0 | 0 | 0 | 0 | 0 |
| | | 45˚ | 0 | 0 | 0 | 5 | 5 | 15 |
| | | 60˚ | 0 | 0 | 0 | 50 | 45 | 70 |
| | 6–7 | 0˚ | 15 | 0 | 0 | 0 | 0 | 0 |
| | | 30˚ | 15 | 25 | 0 | 0 | 0 | 0 |
| | | 45˚ | 10 | 20 | 5 | 0 | 0 | 0 |
| | | 60˚ | 15 | 10 | 15 | 10 | 10 | 15 |
| Mn | 5–6 | 0˚ | 0 | 0 | 0 | 0 | 0 | 0 |
| | | 30˚ | 0 | 0 | 0 | 0 | 0 | 0 |
| | | 45˚ | 0 | 0 | 0 | 0 | 5 | 0 |
| | | 60˚ | 0 | 0 | 0 | 0 | 20 | 55 |
| | 6–7 | 0˚ | 0 | 0 | 0 | 0 | 0 | 5 |
| | | 30˚ | 0 | 0 | 0 | 0 | 0 | 5 |
| | | 45˚ | 0 | 0 | 0 | 0 | 0 | 5 |
| | | 60˚ | 0 | 0 | 0 | 5 | 5 | 15 |

Mx, maxilla; Mn, mandible; 5–6, interradicular site between second premolar and first molar; 6–7, interradicular site between first and second molars; hyper, hyperdivergent; normo, normodivergent; hypo, hypodivergent.

Contact group: The miniscrew directly contacts the root upon miniscrew insertion.

Noncontact group: The miniscrew does not interfere with root movement because it is located buccally outside the root distalization path.

**Table 6. Results of generalized estimating equations.**

| | P | |
|---|---|---|
| Facial type(F) | .021 | Hypo>Normo>Hyper |
| Jaw(J) | <.001 | Mn>Mx |
| Interradicular Site(IS) | <.001 | 5–6>6–7 |
| Insertion Angle(IA) | <.001 | 60˚>45˚>30˚>0˚ |
| F*J | .016 | Mn: Hypo>Normo>Hyper |
| F*IS | .785 | |
| F*IA | .339 | |
| J*IS | <.001 | Mx: 5–6>6–7 |
| | | Mn: 5–6>6–7 |
| J*IA | .001 | Mx: 60˚>45˚>30˚>0˚ |
| | | Mn: 60˚>45˚>30˚>0˚ |
| IS*IA | <.001 | 5–6: 60˚>45˚>30˚>0˚ |
| | | 6–7: 60˚>45˚>30˚>0˚ |
| Sex | .054 | |
| Mandibular plane angle | .450 | |
| Facial height index | .538 | |

Mx, maxilla; Mn, mandible; 5–6, interradicular site between second premolar and first molar; 6–7, interradicular site between first and second molars; Hyper, hyperdivergent; Normo, normodivergent; Hypo, hypodivergent.

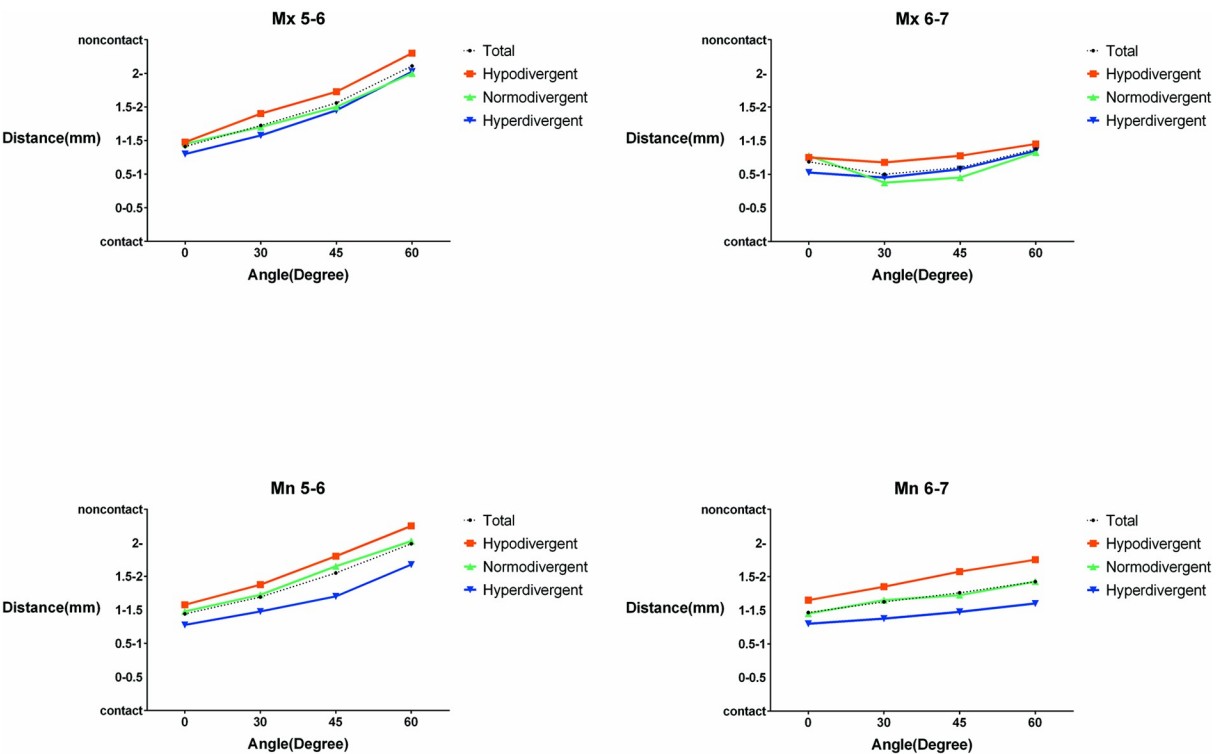

**Fig 6. Pictorial representation of miniscrew–root distances.** Graphs of mean values of categorical miniscrew–root distances according to miniscrew insertion angles at the Mx 5–6, Mx 6–7, Mn 5–6, and Mn 6–7 regions. Mx, maxilla; Mn, mandible; 5–6, interradicular site between second premolar and first molar; 6–7, interradicular site between first and second molars.

hyperdivergent to normodivergent, and then to hypodivergent. However, in the maxilla, there were no significant differences among vertical facial types (Table 6).

Graphs showing the change in mean values of the categorized miniscrew–root distances according to the miniscrew insertion angles at each placement site are shown in Fig 6.

## Discussion

Previous studies have suggested that the appropriate positions for miniscrew placement can be determined by measuring the mesiodistal distance between the roots [5, 6, 23]. However, since the three-dimensional relationship between the miniscrews and roots is determined by the miniscrew insertion angle, diameter, length, and buccal bone thickness, it is difficult to estimate the miniscrew–root distance simply by determining the two-dimensional interradicular distances. Therefore, in this study, the miniscrew–root distances were assessed according to the miniscrew placement angle by a simulated placement of miniscrews with a regular shape and size. Furthermore, the differences in the miniscrew–root distance among hyperdivergent, normodivergent, and hypodivergent facial types were evaluated.

In our study, the miniscrew–root distances increased as the miniscrew placement angle increased in the Mx 5–6, Mn 5–6, and Mn 6–7 regions, and decreased in the Mx 6–7 region. This is because the interradicular distance increases from the cervical area to the apex in the Mx 5–6, Mn 5–6, and Mn 6–7 regions, as suggested in previous studies [5, 10]. Park et al. [10] reported that in the Mx 6–7 region, unlike other molar areas, the interradicular space decreased from the cervical area to the middle part of the root and increased thereafter to the

apex. In our study, the miniscrew–root distance in the Mx 6–7 region decreased when the miniscrew was placed at 30˚ angulation rather than 0˚ and increased when it was placed at 45˚ and 60˚ insertion angles. This indicated that the miniscrews are close to the middle part of the root at a 30˚ insertion angle.

The miniscrew–root distance was greater in the interradicular site between the second premolar and first molar than that in the site between the first and second molars, which could be attributed to the shape of the roots. As the roots of premolars are conical in shape and those of the first and second molars are mostly divergent, the interdental space between the second premolar and first molar is larger toward the middle and apical portions. Our study showed that the rate of miniscrew-root distances > 2 mm of Mx 5–6 and Mn 5–6 regions was greater than that of Mx 6–7 and Mn 6–7 regions when the miniscrews were placed at angles of 60˚ (Table 4). Therefore, it is more advantageous to place a miniscrew for molar distalization in the interradicular space between the second premolar and first molar than in the site between the first and second molars.

The Mx 6–7 region has traditionally been considered an inappropriate position for miniscrew placement because it offers an interradicular space of < 3 mm [5, 6]. Recent three-dimensional studies using CBCT have suggested that Mx 6–7 is the most ideal and safest zone for the placement of miniscrews for maxillary molar distalization because of the presence of thicker buccal bone in the Mx 6–7 region [24, 25]. However, in our study, which simulated miniscrew placement at 4 mm from the CEJ, the miniscrew–root distance in the Mx 6–7 was determined to be smaller than that in the Mx 5–6. Since Liu et al. [24] measured the buccal alveolar bone thickness at a distance of 5 mm above the alveolar crest and the thickest buccal bone was observed at the distance of 11 mm, the level of measurement was placed vertically higher than that used in our study. At a distance of 4 mm from the CEJ, the thickness of buccal bone was not adequate, and the miniscrew–root distance seems to be affected by the narrow interdental space. Since the height of attached gingiva ranges from 4.3 to 5.4 mm [26], it is important to consider that placing the miniscrews above the attached gingiva will cause soft tissue irritation and gingival inflammation, which could result in miniscrew failure [27]. In addition, our study showed that root contact occurred only in the Mx 6–7 region (Table 5). Kuroda et al. [3] reported that root proximity was a major factor for miniscrew failure, and it is known that root contact by miniscrews can cause external root resorption [28]. Thus, we suggest that miniscrews should be placed in the Mx 5–6 region rather than Mx 6–7, within the range of attached gingiva, because the miniscrew–root distance is wider, and the probability of root contact is decreased.

A change in the miniscrew insertion angle alters the bone penetration depth of the miniscrew and the distance between the miniscrew and the root in the buccolingual axis. The present evidence indicates that the thickness of alveolar bone may affect the miniscrew–root distance in the buccolingual axis. Previously, cortical bone thickness, cortical bone density, and alveolar bone thickness were found to be related to facial divergence [11, 29, 30]. Horner et al. [11] assessed the relationship between vertical facial type and bone characteristics such as cortical bone thickness and total alveolar ridge thickness using computed tomography in the maxilla and mandible. They found that in the mandibular posterior buccal area, hypodivergent subjects showed increased cortical bone thicknesses and total alveolar ridge thicknesses than hyperdivergent subjects. However, in the maxillary posterior region, the difference among vertical facial types was not as prominent as in the mandible, and this difference was not statistically significant. In this study, the results of GEE were similar to those reported by Horner et al. [11] When interactions between vertical facial types and the jaws were analyzed, significant differences among vertical facial types in the mandible, but not in the maxilla, were revealed.

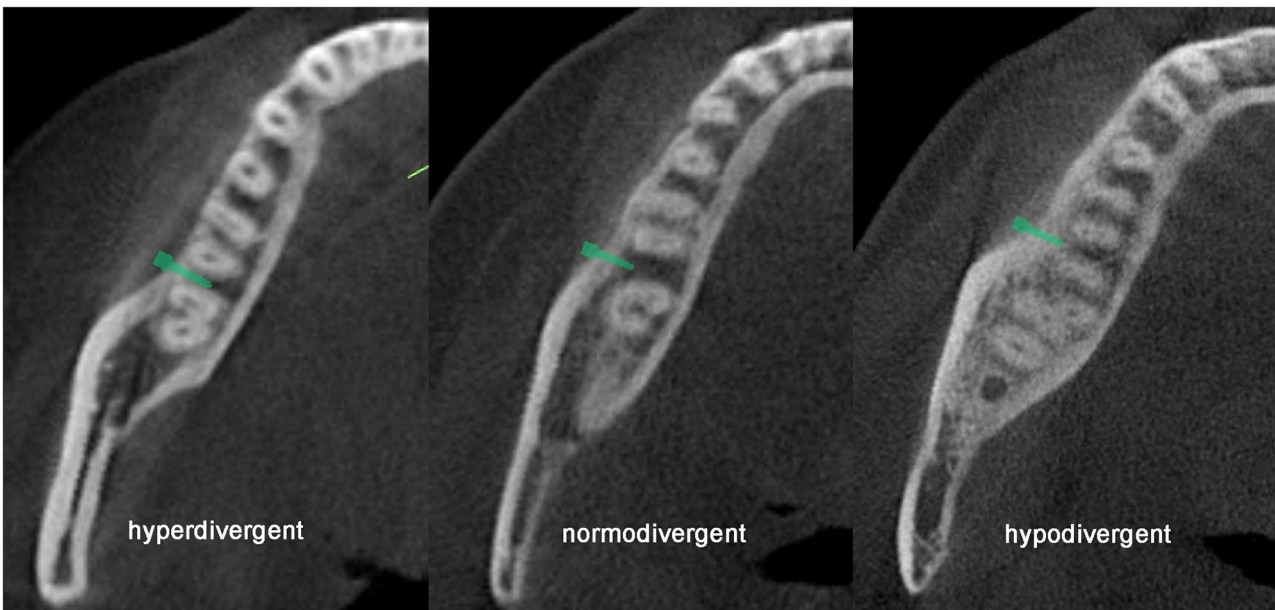

**Fig 7. Miniscrew–root distances in the three vertical facial types.** Different relationships between roots and miniscrews in the three vertical facial types in the axial slice, demonstrating examples of hyperdivergent, normodivergent, and hypodivergent facial types.

The difference among vertical facial types was more pronounced in the mandible than that in the maxilla because of the influence of the masticatory muscles [31]. According to the Wolff's law, if the load on a particular bone increases, the bone remodels itself to resist this increase [32]. In terms of functional anatomy, the mandibular area closest to the ramus, where the masticatory muscles are attached, resists forces applied from a buccal direction [29, 30]. Multiple studies have reported that larger masticatory muscle forces are associated with a wider ramus, more rectangular body, flat mandibular plane, small gonial angle, and greater posterior facial height [31, 33]. When comparing vertical facial types, differences in masticatory muscle forces lead to differences in bone morphology around the mandibular molar region [34], resulting in differences in the distance between the miniscrew and root surface (Fig 7).

An interesting finding of this study was that the observed rates of the noncontact group were up to 70% and 55% when the miniscrews were placed at 60° insertion angles in the Mx 5–6 and Mn 5–6 regions, respectively (Table 5). In this study, the shortest distance between the root and miniscrew was measured. The posterior occlusal line was used as a reference to measure the miniscrew–root distance for molar distalization, because the molars would be distalized along the posterior occlusal line. The miniscrew–root distance was measured on a line parallel to the reference line. Therefore, if the angulated miniscrew was placed buccally outside the molar distalization path, the miniscrew–root distance could not be measured, and such cases were categorized in the noncontact group. A clinical case study on molar distalization using miniscrews has reported that miniscrews did not fail even when the miniscrew, initially located between the roots, was located directly on the buccal side of the anterior root as the molars moved distally [35]. The reason could be that angulated miniscrews are located outside the root distalization path and do not interfere with root movement. In such cases, the orthodontist can perform extensive molar distalization without the need for relocating the miniscrew, thus increasing the treatment efficiency.

When the periodontal ligament (PDL) was damaged by the miniscrew, extensive root resorption was observed if the miniscrew was not removed immediately [4, 28]. In addition, Kim et al. [28] reported that external root resorption occurs even after the miniscrew is placed < 1 mm from the PDL. This is reported to be due to compressive stresses in PDL activating osteoclastogenesis within the PDL [36]. Therefore, Maino et al. [37] recommended a 1 mm clearance between the miniscrew and the root for periodontal health and miniscrew stability. However, in computed tomography images, the surface boundary of the lamina dura is difficult to identify, and the root surface is more clearly visible. Therefore, in this study, for the accuracy and reproducibility of the measurement, miniscrew-root distance was measured from the mesial surface of the miniscrew to the distal surface of the root of the anterior tooth.

In this study, the maximum angulation was regarded as 60˚, which is in contrast to the more radical vertical angles suggested by other studies. Previous studies have reported that increasingly oblique placement of miniscrews increases the cortical bone contact and placement torque, which positively affects their stability [13, 22]. However, more oblique placement of miniscrews has also been shown to increase bone stresses around the miniscrew and create a longer lever arm, which decreases anchorage resistance [15]. In particular, if the miniscrew placement angle is excessively steep, miniscrew slippage or maxillary sinus perforation can occur [38]. Therefore, excessive miniscrew angulation is not recommended.

Despite the strengths of virtual simulation, this study also had some limitations that warrant discussion. First, it is difficult to accurately predict the amount of possible molar distalization using the results of this study. Since distal movement of molars is accompanied by molar tipping, the amount of possible molar distalization will be greater than the miniscrew–root distance measured at the root level [2]. In addition, molar distalization can be limited by anatomical structures such as maxillary tuberosity or mandibular lingual cortex even if the root does not come in contact with the miniscrew [39]. Second, as it is difficult to delineate soft tissues in CBCT [40], soft tissues were not considered in this study, and the miniscrew head was placed directly on the bone surface. Placement of the miniscrews at certain distances from the cortical bone to compensate for the soft tissue thickness was considered, but it was less reproducible in several measurements. Third, in practice, optimal placement of a miniscrew in the midpoint between the roots at the desired insertion angle is difficult. Surgical guides developed using digital model imaging can be used for more accurate placement of the miniscrews [41]. Considering these points, further clinical studies with more refined designs are required to provide guidance for clinical practice.

## Conclusion

The miniscrew–root distance was greater in the mandible than that in the maxilla. An increase in the miniscrew insertion angle was found to significantly increase the miniscrew–root distance, and the amount of increase was affected by the miniscrew placement site and vertical facial type. The effect of a vertical facial type was significant in the mandible, with the greatest miniscrew–root distance in the hypodivergent facial type, followed by the normodivergent facial type and the hyperdivergent facial type. However. this effect was not evident in the maxilla. Thus, placement of miniscrews in the interradicular site between the second premolar and first molar, rather than between the first and second molars, was found to be advantageous for molar distalization.

## Author Contributions

**Conceptualization:** Ju-Hyun Yoon, Jung-Yul Cha, Yoon Jeong Choi, Won-Se Park, Sang-Sun Han, Kee-Joon Lee.

**Data curation:** Ju-Hyun Yoon, Kee-Joon Lee.

**Formal analysis:** Ju-Hyun Yoon, Jung-Yul Cha, Yoon Jeong Choi, Won-Se Park, Sang-Sun Han, Kee-Joon Lee.

**Funding acquisition:** Kee-Joon Lee.

**Investigation:** Ju-Hyun Yoon, Kee-Joon Lee.

**Methodology:** Ju-Hyun Yoon, Kee-Joon Lee.

**Project administration:** Kee-Joon Lee.

**Resources:** Kee-Joon Lee.

**Software:** Kee-Joon Lee.

**Supervision:** Jung-Yul Cha, Yoon Jeong Choi, Won-Se Park, Sang-Sun Han, Kee-Joon Lee.

**Validation:** Jung-Yul Cha, Yoon Jeong Choi, Won-Se Park, Sang-Sun Han, Kee-Joon Lee.

**Visualization:** Kee-Joon Lee.

**Writing – original draft:** Ju-Hyun Yoon, Kee-Joon Lee.

**Writing – review & editing:** Ju-Hyun Yoon, Kee-Joon Lee.

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
