## [Decision Letter · Decision Letter 0]

31 Mar 2020

PONE-D-20-04649

Simulation of miniscrew-root distance in molar distalization depending on miniscrew insertion angle and vertical facial types.

PLOS ONE

Dear Dr lee,

Thank you for submitting your manuscript to PLOS ONE. After careful consideration, we feel that it has merit but does not fully meet PLOS ONE’s publication criteria as it currently stands. Therefore, we invite you to submit a revised version of the manuscript that addresses the points raised during the review process.

Some minor revision has been suggested by both reviewers. Please make the necessary changes.

We would appreciate receiving your revised manuscript by May 15 2020 11:59PM. To enhance the reproducibility of your results, we recommend that if applicable you deposit your laboratory protocols in protocols.io, where a protocol can be assigned its own identifier (DOI) such that it can be cited independently in the future. For instructions see: http://journals.plos.org/plosone/s/submission-guidelines#loc-laboratory-protocols

We look forward to receiving your revised manuscript.

Kind regards,

Claudia Trindade Mattos, Ph.D.

Academic Editor

PLOS ONE

Journal Requirements:

Reviewers' comments:

Reviewer's Responses to Questions

**Comments to the Author**

1. Is the manuscript technically sound, and do the data support the conclusions?

Reviewer #1: Partly

Reviewer #2: Yes

2. Has the statistical analysis been performed appropriately and rigorously? 

Reviewer #1: Yes

Reviewer #2: Yes

3. Have the authors made all data underlying the findings in their manuscript fully available?

Reviewer #1: Yes

Reviewer #2: Yes

4. Is the manuscript presented in an intelligible fashion and written in standard English?

Reviewer #1: Yes

Reviewer #2: Yes

5. Review Comments to the Author

Reviewer #1: Question 1: Partly. A sample size estimation should be reported.

Question 2: Yes.

Question 3: Yes.

Question 4: Yes. Despite this manuscript was written in standard English, a language editing service is recommended.

This was a retrospective, cross-sectional study that evaluated the effects of miniscrew insertion angle and vertical facial type on miniscrew–root distances in molar distalization in the maxilla and mandible via simulated placement of interradicular miniscrews in patients’ cone-beam computed tomography (CBCT) images. The investigation of miniscrew-root distance is a clinically relevant topic due to the versatility provided by temporary anchorage devices on orthodontic biomechanics. The study was well designed and the manuscript was well written. There is however, some revision needed.

This manuscript should be revised by a professional English language editing service.

TITLE

“Simulation of miniscrew-root distance in molar distalization depending on miniscrew insertion angle and vertical facial types.”

The terms “molar distalization” should be revised in the Title, since that, in fact, the miniscrew-root distance was evaluated based on a virtual simulation of a miniscrew placement. Since the virtual simulation analysis did not comprise a prediction of moving posterior teeth distally, it should be clear that “molar distalization” refers to a clinical application of the evidence provided by this study.

ABSTRACT

The minimum and maximum miniscrew-root distance values regarding insertion angle and facial type should be reported in the Abstract.

MATERIAL AND METHODS

- How was this study sample size estimated? Was it based on a previous published or a pilot study? The criteria adopted for sample size calculation should be reported.

RESULTS

- The minimum and maximum miniscrew-root distance values regarding insertion angle and facial type should be reported in the following sentences:

“At each miniscrew placement site, the minimum and maximum miniscrew–root

distances were measured at 0° insertion angle in hyperdivergent facial type and 60° insertion angle in hypodivergent facial type, respectively. Exceptionally, in the Mx 6-7 region, the minimum distance was measured at 30° insertion angle in hyperdivergent facial type (Table 3)”.

DISCUSSION

- The rates of miniscrew-root distances > 2 mm of miniscrews placed at angles of 60o (Table 6) should be reported in Results, rather than in Discussion section.

- The effect of miniscrew placement on adjacent periodontal ligament should be discussed, especially because, in the present study, the miniscrew-root distance was measured as the shortest linear distance from the mesial surface of the miniscrew to the distal surface of the root of the anterior tooth.

CONCLUSION

- The results regarding the influence of the jaw (Mn>Mx) on the miniscrew-root distances should also be reported in the Conclusion.

Reviewer #2: This is an interesting and well written investigation. There are only minor improvements to be performed.

The tables, figures and figure captions should be displayed at the end of the manuscript.

The conclusions should be summarized.

6. PLOS authors have the option to publish the peer review history of their article (what does this mean?). If published, this will include your full peer review and any attached files.

Reviewer #1: No

Reviewer #2: No

---

## [Author Response · Author response to Decision Letter 0]

24 May 2020

Responses to the Reviewers’ Comments

Manuscript number: PONE-D-20-04649

Title: Simulation of miniscrew-root distance in molar distalization depending on miniscrew insertion angle and vertical facial types

Reviewer #1:

This was a retrospective, cross-sectional study that evaluated the effects of miniscrew insertion angle and vertical facial type on miniscrew–root distances in molar distalization in the maxilla and mandible via simulated placement of interradicular miniscrews in patients’ cone-beam computed tomography (CBCT) images. The investigation of miniscrew-root distance is a clinically relevant topic due to the versatility provided by temporary anchorage devices on orthodontic biomechanics. The study was well designed and the manuscript was well written. There is however, some revision needed.

Comment 1: This manuscript should be revised by a professional English language editing service.

Response: Thank you for your helpful suggestion. We apologize for the errors in language and grammar, even though the manuscript was checked several times by English editing processes prior to submission. In this revised version, the manuscript has been checked by a native English speaker from an English editing company once again to ensure that the language and grammar are correct, and the manuscript has been revised as a whole. We have attached the certificate of English editing.

Comment 2: TITLE

“Simulation of miniscrew-root distance in molar distalization depending on miniscrew insertion angle and vertical facial types.”

The terms “molar distalization” should be revised in the Title, since that, in fact, the miniscrew-root distance was evaluated based on a virtual simulation of a miniscrew placement. Since the virtual simulation analysis did not comprise a prediction of moving posterior teeth distally, it should be clear that “molar distalization” refers to a clinical application of the evidence provided by this study.

Response: Thank you for your comment. We agree with your recommendation. We have revised the term “molar distalization” to “possible molar distalization” in the Title. Also, we have revised the term throughout the revised manuscript.

As described in the Discussion section of our manuscript, the measured miniscrew-root distance could not guarantee the molar disalization distance. This is because it is impossible to predict what can happen in the actual distal movement of molars, such as the root touching the anatomical structure before the miniscrew and root contact, in virtual simulation analysis.

In this study, under the premise of distal molar movement, the shortest linear distance between the anterior root and miniscrew was measured, and we have tried to determine the “possible” molar distalization distance despite several limitations. Therefore, we think it is appropriate to revise the term “molar distalization” to “possible molar distalization”. We appreciate your comment.

Revision:

The title was revised.

“Simulation of miniscrew-root distance for possible molar distalization depending on miniscrew insertion angle and vertical facial types.”

Comment 3: ABSTRACT

The minimum and maximum miniscrew-root distance values regarding insertion angle and facial type should be reported in the Abstract.

Response: Thank you for your comment. We have added the minimum and maximum mean of the measurable miniscrew-root distance regarding each insertion angle and facial type in the Abstract section in the revised manuscript (page 2, line 45-47). Because the Abstract section is recommended not to exceed 300 words by the submission guidelines, we have removed one previous sentence in the revised manuscript. Also, instead of reporting the minimum and maximum mean of the measurable miniscrew-root distance at each of the miniscrew placement sites, the most minimum and maximum values were reported in the revised Abstract. We appreciate your comment.

Revision:

The Abstract section was revised, and added sentence is presented.

“The minimum mean distance was found in the Mx 6-7 (30°; 0.86±0.35 mm), and the maximum mean distance was found in the Mn 5-6 (60°; 2.64±0.56 mm).”

Comment 4: MATERIAL AND METHODS

 How was this study sample size estimated? Was it based on a previous published or a pilot study? The criteria adopted for sample size calculation should be reported.

Response: Thank you for your comment on a very important part that we did not describe. Our study on the effects of the miniscrew insertion angle and vertical facial type on miniscrew–root distances in molar distalization is an explorative pilot study that has not been studied before. Therefore, the number of subjects for each group was determined to be as similar as possible; the number for each group was set to 20, which was based on the sample numbers suggested as suitable for the pilot study in the reference studies. The criteria is presented in the revised manuscript (page 6, line 144-145). We appreciate your comment.

References

Julious, S. A. (2005). Sample size of 12 per group rule of thumb for a pilot study. Pharmaceutical Statistics, 4, 287-291.

Isaac, S., & Michael, W. B. (1995). Handbook in research and evaluation. San Diego, CA: Educational and Industrial Testing Services

van Belle, G., 2008. Sample Size. In Statistical Rules of Thumb. Wiley, Chichester. pp. 27–51.

Revision:

The Materials and methods section was revised, and we added the following sentence:

“Since our study is an explorative pilot study, 20 subjects for each group were determined in consideration of sample numbers suggested as suitable for the pilot study.”

Comment 5: RESULTS

 The minimum and maximum miniscrew-root distance values regarding insertion angle and facial type should be reported in the following sentences:

“At each miniscrew placement site, the minimum and maximum miniscrew–root

distances were measured at 0° insertion angle in hyperdivergent facial type and 60° insertion angle in hypodivergent facial type, respectively. Exceptionally, in the Mx 6-7 region, the minimum distance was measured at 30° insertion angle in hyperdivergent facial type (Table 3)”.

Response: Thank you for your comment. We have added the minimum and maximum means of the measurable miniscrew-root distance to the Results section in the revised manuscript (page 12, line 255-264). We appreciate your comment.

Revision:

The Results section was revised, and the revised paragraph was presented.

“At Mx 5-6, Mn 5-6, and Mn 6-7 miniscrew placement sites, the minimum mean of the measurable miniscrew-root distance (Mx 5-6: 1.08±0.25 mm; Mn 5-6: 1.04±0.14 mm; Mn 6-7: 1.03±0.25 mm) was measured at a 0° insertion angle in hyperdivergent facial types, and the maximum mean of the measurable miniscrew-root distance (Mx 5-6: 2.16±0.33 mm; Mn 5-6: 2.64±0.56 mm; Mn 6-7: 2.05±0.61 mm) was measured at a 60° insertion angle in hypodivergent facial types. In exception, in the Mx 6-7 region, the minimum mean of the measurable miniscrew-root distance (Mx 6-7: 0.86±0.35 mm) was measured at a 30° insertion angle in hyperdivergent facial types, and the maximum mean of the measurable miniscrew-root distance (Mx 6-7: 1.17±0.48 mm) was measured at a 60° insertion angle in hypodivergent facial types (Table 3).”

Comment 6: DISCUSSION

- The rates of miniscrew-root distances > 2 mm of miniscrews placed at angles of 60˚ (Table 6) should be reported in Results, rather than in Discussion section.

Response: Thank you for your encouraging comments and helpful suggestion. We agree with your recommendation. Table 6 was moved from the Discussion section to the Results section, and Table 6 was renamed to Table 4 (page 13, line 272-278). The existing Tables 4 and 5 were changed to Tables 5 and 6. We appreciate your comment.

Comment 7: DISCUSSION

- The effect of miniscrew placement on adjacent periodontal ligament should be discussed, especially because, in the present study, the miniscrew-root distance was measured as the shortest linear distance from the mesial surface of the miniscrew to the distal surface of the root of the anterior tooth.

Response: Thank you for your valuable suggestion. In accordance with your suggestion, we have added a paragraph on the effect of miniscrew placement on the adjacent periodontal ligament in the Discussion section (page 21, line 422-433) and have provided references for that in the revised manuscript (page 26-27, line 558-563). We appreciate your comment.

Revision:

The Discussion section was revised, and the following paragraph was added:

“When the periodontal ligament (PDL) was damaged by the miniscrew, extensive root resorption was observed if the miniscrew was not removed immediately [4, 25]. In addition, Kim et al [25] reported that external root resorption occurs even after the miniscrew is placed < 1 mm from the PDL. This is reported to be due to compressive stresses in PDL activating osteoclastogenesis within the PDL [33]. Therefore, Maino et al [34] recommended a 1 mm clearance between the miniscrew and the root for periodontal health and miniscrew stability. Taking this into consideration, when measuring the miniscrew-root distance in this study, the distance from the lamina dura to the miniscrew surface can be measured. However, in computed tomography images, the surface boundary of the lamina dura is difficult to identify, and the root surface is more clearly visible. Therefore, in this study, for the accuracy and reproducibility of the measurement, miniscrew-root distance was measured from the mesial surface of the miniscrew to the distal surface of the root of the anterior tooth.”

Comment 8: CONCLUSION

- The results regarding the influence of the jaw (Mn>Mx) on the miniscrew-root distances should also be reported in the Conclusion.

Response: Thank you for your comment. We agree with your recommendation. We have added a sentence regarding the influence of the jaw on the miniscrew-root distance in the Discussion section of the revised manuscript (page 22, line 459). We appreciate your comment.

Revision:

The Conclusion section was revised and the following sentence added:

“The miniscrew–root distance was greater in the mandible than that in the maxilla.”

Reviewer #2:

 This is an interesting and well written investigation. There are only minor improvements to be performed.

Comment 1: The tables, figures and figure captions should be displayed at the end of the manuscript.

Response: Thank you for your kind comment. According to the submission guidelines of Plos one, the tables and figure captions are recommended to appear directly after the paragraph in which they are first cited. Therefore, we have placed the tables and figure captions in the middle of the manuscript to be included directly after the paragraph in which they are first cited. Figures were uploaded separately as individual files. We appreciate your comment.

Comment 2: The conclusions should be summarized.

Response: Thank you for your insightful comment. We agree with your recommendation. As your suggestion, we have tried to remove unnecessary sentences and leave only the important contents. We summarized the four key points in this study (jaw, miniscrew placement angle, vertical facial type, and placement site) in the Conclusion section of the revised manuscript (page 22-23, line 459-467). We appreciate your comment.

Revision:

The Conclusion section was revised and presented.

“The miniscrew–root distance was greater in the mandible than that in the maxilla. In the Mx 5-6, Mn 5-6, and Mn 6-7 regions, but not in the Mx 6-7 region, an increase in the miniscrew insertion angle was found to significantly increase the miniscrew–root distance. The effect of a vertical facial type was significant in the mandible, with the greatest miniscrew–root distance in the hypodivergent facial type, followed by the normodivergent facial type and the hyperdivergent facial type. However. this effect was not evident in the maxilla. Thus, placement of miniscrews in the interradicular site between the second premolar and first molar, rather than between the first and second molars, was found to be advantageous for molar distalization.”

---

## [Decision Letter · Decision Letter 1]

22 Jul 2020

PONE-D-20-04649R1

Simulation of miniscrew-root distance for possible molar distalization depending on miniscrew insertion angle and vertical facial types.

PLOS ONE

Dear Dr. lee,

Thank you for submitting your manuscript to PLOS ONE. After careful consideration, we feel that it has merit but does not fully meet PLOS ONE’s publication criteria as it currently stands. Therefore, we invite you to submit a revised version of the manuscript that addresses the points raised during the review process.

There are some minor revision needed appointed by one of the reviewers. We are looking forward to your revised manuscript.

We look forward to receiving your revised manuscript.

Kind regards,

Claudia Trindade Mattos, Ph.D.

Academic Editor

PLOS ONE

Reviewers' comments:

Reviewer's Responses to Questions

**Comments to the Author**

1. If the authors have adequately addressed your comments raised in a previous round of review and you feel that this manuscript is now acceptable for publication, you may indicate that here to bypass the “Comments to the Author” section, enter your conflict of interest statement in the “Confidential to Editor” section, and submit your "Accept" recommendation.

Reviewer #1: (No Response)

Reviewer #2: All comments have been addressed

2. Is the manuscript technically sound, and do the data support the conclusions?

Reviewer #1: Partly

Reviewer #2: Yes

3. Has the statistical analysis been performed appropriately and rigorously? 

Reviewer #1: Yes

Reviewer #2: Yes

4. Have the authors made all data underlying the findings in their manuscript fully available?

Reviewer #1: Yes

Reviewer #2: Yes

5. Is the manuscript presented in an intelligible fashion and written in standard English?

Reviewer #1: Yes

Reviewer #2: Yes

6. Review Comments to the Author

Reviewer #1: Reviewers' requests and comments have been successfully addressed and the manuscript was improved.

I have some remaining minor points:

TITLE

A suggestion for the title: “Simulation of miniscrew-root distance available for molar distalization depending on miniscrew insertion angle and vertical facial types.”

MATERIAL AND METHODS

Please provide the reference studies citation following the sentence referring to study sample size (page 6 lines 144-145).

DISCUSSION

As a suggestion, the following sentence should be removed: “Taking this into consideration, when measuring the miniscrew-root distance in this study, the distance from the lamina dura to the miniscrew surface can be measured.”, since it is reported in the following sentence that the lamina dura is a difficult structure to be measured in CBCTs (page 21, lines 428-429).

CONCLUSION

Please check the sentence: “In the Mx 5-6, Mn 5-6, and Mn 6-7 regions, but not in the Mx 6-7 region, an increase in the miniscrew insertion angle was found to significantly increase the miniscrew–root distance.” (page 22, lines 459-461) According to the results of GEE analysis presented in Table 6, the miniscrew–root distance increased as the miniscrew placement angle increased (p<.001). This result doesn’t mention any exception, so the following statement should be checked: “but not in the Mx 6-7 region” and “except in the Mx 6-7 region” in the conclusions of the manuscript and the abstract, respectively.

Reviewer #2: The authors have satisfactorily addressed all my concerns and therefore I recommend publication of this manuscript.

7. PLOS authors have the option to publish the peer review history of their article (what does this mean?). If published, this will include your full peer review and any attached files.

Reviewer #1: No

Reviewer #2: No

---

## [Author Response · Author response to Decision Letter 1]

18 Aug 2020

Responses to the Reviewers’ Comments

Manuscript number: PONE-D-20-04649R1

Title: Simulation of miniscrew-root distance for possible molar distalization depending on miniscrew insertion angle and vertical facial types

Reviewer #1:

Reviewers' requests and comments have been successfully addressed and the manuscript was improved.

I have some remaining minor points:

Comment 1: TITLE

A suggestion for the title: “Simulation of miniscrew-root distance available for molar distalization depending on miniscrew insertion angle and vertical facial types.”

Response:

Thank you for your helpful suggestion. We have now revised the title according to your suggestion, as well as the corresponding parts in the revised manuscript. We appreciate your comment, as we believe your input has improved our submission.

Revision:

The title was revised.

Before:

“Simulation of miniscrew-root distance for possible molar distalization depending on miniscrew insertion angle and vertical facial types.”

After:

“Simulation of miniscrew-root distance available for molar distalization depending on the miniscrew insertion angle and vertical facial type.”

Comment 2: MATERIAL AND METHODS

Please provide the reference studies citation following the sentence referring to study sample size (page 6 lines 144-145).

Response:

Thank you for your comment and we apologize for this issue. As requested, we have now added the citations for the reference studies.

(page 6, lines 141-143)

Since our study is an explorative pilot study, 20 subjects for each group were determined in consideration of sample numbers suggested as suitable for the pilot study [19-21].

(page 24, lines 509-514)

References

19. Julious SA. Sample size of 12 per group rule of thumb for a pilot study. Pharmaceutical Statistics: The Journal of Applied Statistics in the Pharmaceutical Industry. 2005;4(4):287-91.

20. Isaac S, MICHAEL W. Handbook in Research and Evaluation. San Diego, Californ4ia: ED. ITS Publishers; 1971.

21. Van Belle G. Sample size. Wiley Hoboken (NJ); 2008. p. 27-52.

Comment 3: DISCUSSION

As a suggestion, the following sentence should be removed: “Taking this into consideration, when measuring the miniscrew-root distance in this study, the distance from the lamina dura to the miniscrew surface can be measured.”, since it is reported in the following sentence that the lamina dura is a difficult structure to be measured in CBCTs (page 21, lines 428-429).

Response:

We appreciate your insightful comment. We agree with your suggestion and have now removed the sentence in the revised manuscript.

Comment 4: CONCLUSION

Please check the sentence: “In the Mx 5-6, Mn 5-6, and Mn 6-7 regions, but not in the Mx 6-7 region, an increase in the miniscrew insertion angle was found to significantly increase the miniscrew–root distance.” (page 22, lines 459-461) According to the results of GEE analysis presented in Table 6, the miniscrew–root distance increased as the miniscrew placement angle increased (p<.001). This result doesn’t mention any exception, so the following statement should be checked: “but not in the Mx 6-7 region” and “except in the Mx 6-7 region” in the conclusions of the manuscript and the abstract, respectively.

Response:

We would like to thank you for your suggestion, and we agree with your comments. In the abstract and conclusions of the manuscript, we aimed to describe that the pattern of increasing miniscrew-root distance as the miniscrew placement angle increases varies according to the miniscrew placement site and facial type (Table 3, Figure 6, GEE results in Table 6). However, as you have commented, the sentences were unclear and contradictory. The phrases “but not in the Mx 6-7 region” and “except in the Mx 6-7 region” have now been deleted from the conclusions of the manuscript and the abstract, respectively, and the remaining statements have been revised.

Revision:

The abstract was revised.

Before:

Miniscrew–root distance increased as the insertion angle increased from 0° to 60° at all placement sites, except in the Mx 6-7 region.

After:

Miniscrew–root distance significantly increased as the insertion angle increased from 0° to 60°.

(page 2, lines 42-43)

Before:

Miniscrew–root distance increased with the increased insertion angle in all posterior interradicular areas, except in the Mx 6-7 region and in the hypodivergent facial type.

After:

Miniscrew–root distance increased significantly with the increased insertion angle, and the amount of increase was affected by the miniscrew placement site and vertical facial type.

(page 2, lines 49-50; page 3, line 51)

The conclusion was revised.

Before:

In the Mx 5-6, Mn 5-6, and Mn 6-7 regions, but not in the Mx 6-7 region, an increase in the miniscrew insertion angle was found to significantly increase the miniscrew–root distance.

After:

An increase in the miniscrew insertion angle was found to significantly increase the miniscrew–root distance, and the amount of increase was affected by the miniscrew placement site and vertical facial type.

(page 22, lines 452-455)

Reviewer #2:

The authors have satisfactorily addressed all my concerns and therefore I recommend publication of this manuscript.

---

## [Editor Report · Decision Letter 2]

14 Sep 2020

Simulation of miniscrew-root distance available for molar distalization depending on the miniscrew insertion angle and vertical facial type

PONE-D-20-04649R2

Dear Dr. lee,

We’re pleased to inform you that your manuscript has been judged scientifically suitable for publication and will be formally accepted for publication once it meets all outstanding technical requirements.

Kind regards,

Claudia Trindade Mattos, Ph.D.

Academic Editor

PLOS ONE

Additional Editor Comments (optional):

All reviewers' suggestions have been addressed satisfactorily.
---

## [Editor Report · Acceptance letter]

16 Sep 2020

PONE-D-20-04649R2

Simulation of miniscrew-root distance available for molar distalization depending on the miniscrew insertion angle and vertical facial type

Dear Dr. lee:

I'm pleased to inform you that your manuscript has been deemed suitable for publication in PLOS ONE. Congratulations! Your manuscript is now with our production department.

Kind regards,

on behalf of

Dr. Claudia Trindade Mattos 

Academic Editor

PLOS ONE